# CinePile: A Long Video Question Answering Dataset and Benchmark

Ruchit Rawal ♦     Khalid Saifullah ♦     Ronen Basri ♣

David Jacobs ♦     Gowthami Somepalli⋆♦     Tom Goldstein⋆♦

♦ University of Maryland, College Park     ♣ Weizmann Institute of Science

## Abstract

*Current datasets for long-form video understanding often fall short in providing genuine long-form comprehension challenges, as many tasks derived from these datasets can be successfully tackled by analyzing just one or a few random frames from a video. To address this issue, we present a novel dataset and benchmark, CinePile, specifically designed for authentic long-form video understanding. This paper details our innovative approach for creating a question-answer dataset, utilizing advanced LLMs and building upon human-generated raw data. Our comprehensive dataset comprises 200,000 multiple-choice questions (MCQs), covering a diverse range of visual and multimodal aspects, including temporal comprehension, understanding of human-object interactions, and reasoning about events or actions within a scene. Additionally, we evaluate recent advances in video-centric LLMs, both open-source and proprietary, using the evaluation split of our dataset. The findings reveal that even state-of-the-art vision LLMs significantly lag behind human performance in these tasks, highlighting the challenges inherent to video understanding.*

## 1. Introduction

Large multi-modal models offer the potential to analyze and understand long, complex videos. However, training and evaluating them on video data poses difficult challenges. Most videos contain dialog and pixel data, both essential for a complete scene understanding. Furthermore, existing vision-language models are primarily pre-trained on still frames, while understanding long videos requires identifying interactions and plot progressions over time.

In this paper, we introduce CinePile, a large-scale dataset consisting of over 200,000 question-answer pairs from 8000 videos, split into a train and test set. Our dataset emphasizes question diversity, and topics span temporal understanding, perceptual analysis, complex reasoning, and more. It also emphasizes question difficulty, with humans exceeding the best commercial models by approximately 20%, and exceeding open source models by 50%.

We present a scene and a few question-answer pairs from our dataset in Fig. 1. Consider the first question, `How does Gru's emotional state transition throughout the scene?` For a model to answer this correctly, it needs to understand both the visual and temporal aspects, and even reason about the plot progression of the scene. To answer the second question, `What are the objects poking out of the book cover and what is their purpose`, the model must localize an object in time and space, and use its world knowledge to reason about their purpose.

CinePile addresses several weakness of existing video understanding datasets. First, CinePile's large size enables it to serve as both an instruction-tuning dataset and an evaluation benchmark. We believe the ability to do instruction tuning for video at a scale comparable to common language-only instruction datasets will lead to large improvements in model performance. Also, the diversity in CinePile makes it a more comprehensive measure of model performance than existing benchmarks. Unlike existing metrics, CinePile puts little emphasis on purely visual questions (e.g., 'What color is the car'), or on classification questions (e.g., 'What genre is the video') that do not require temporal understanding. Rather, CinePile comprehensively evaluates vision, temporal reasoning, and video understanding while still providing a breakdown of question types to help developers identify blind spots in their models.

CinePile's large size is made possible by our novel pipeline for automated question generation using large language models. Our method leverages large existing sets of movie descriptions created to assist the vision impaired. We transcribe these movie descriptions, and align them with publicly available video clips from YouTube. Using this detailed human analysis of scenes, powerful LLMs are able to create complex and difficult questions without relying to video. At test time, models must address these questions from only the dialog and video frames, and without access to the hand-written descriptions used to build the questions.

---

⋆Equal contribution. Correspondence: ruchitr@umd.edu.

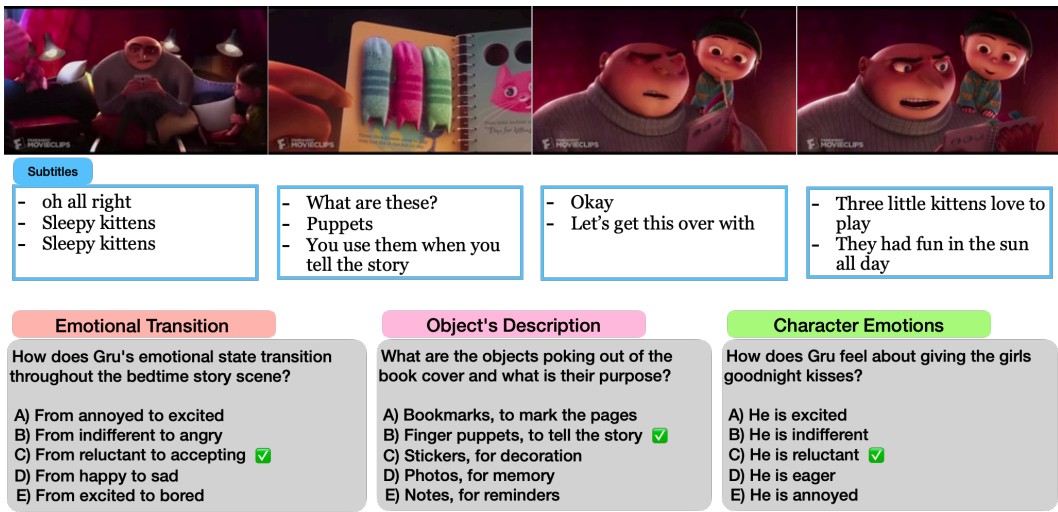

Figure 1. **Example movie clip and multiple-choice questions from CinePile**. The first and second rows depict a selection of image frames extracted from a movie clip from Despicable Me, accompanied by their corresponding subtitles. The next row showcases example questions along with the question template shown in colored headers.

## 2. Creating a long video reasoning benchmark

Our dataset curation process has four primary components 1) Collection of raw video and related data. 2) Generation of question templates. 3) Automated construction of the question-answer dataset using video and templates, and 4) A novel filtering pipeline to remove malformed questions.

### 2.1. Data collection and consolidation

We obtain clips from English-language films from the YouTube channel *MovieClips*[1]. This channel hosts self-contained clips, each encapsulating a major plot point, facilitating the creation of a dataset focused on understanding and reasoning. Next, we collected Audio Descriptions from AudioVault[2]. Lastly, we collect movie information for each scene, such as genre, actors, and main plot from IMDB[3].

**Getting visual descriptions of video for free.** Audio descriptions (ADs) feature a narrator who explains the visual elements crucial to the story during pauses in dialogue. ADs have been created for many films to assist the vision impaired. The key distinction between conventional video caption datasets and ADs lies in the contextual nature of the latter. In ADs, humans emphasize the important visual elements in their narrations, unlike other video caption datasets, which tend to be overly descriptive. We use the audio descriptions as a proxy for visual annotation in the videos for our dataset creation. However, since the video clips we gather are typically 2-3 minutes long, and Audio Descriptions (ADs) cover entire movies, we need to align

---

[1] https://www.youtube.com/@MOVIECLIPS
[2] https://audiovault.net/movies
[3] https://www.imdb.com/

and extract the *clip relevant part* from the AD. We discuss this process in detail in Appendix Sec. C.

### 2.2. Automated Question Templates

Mainstream video question-answering benchmarks were written by human annotators. The question-answer pairs are typically curated in one of two ways: 1) Humans are given complete freedom to ask questions about a given scene [25] 2) Humans focus on specific aspects and are trained or provided with examples of questions, encouraging them to write more questions in a similar style [10, 12, 19, 34].

While we use a template-based approach for question generation, rather than confining to a few predefined themes, we propose an automated method to create question templates from existing human-generated questions. We first cluster 30,000 human-generated questions across multiple existing datasets, then use GPT-4 to discern their underlying themes and generate prototypical questions for each template. We discuss the details of the clustering and template discernment process in Appendix Sec. D. In total, we generate 86 unique templates that we categorize into four high-level categories: Character and Relationship Dynamics (CRD), Narrative and Plot Analysis (NPA), Thematic Exploration (TE) and Setting, and Technical Analysis (STA). For a detailed discussion and example questions from each category, please refer to Appendix Sec. E.

### 2.3. Automated QA generation with LLMs

Before creating questions for a scene, we chose the relevant question templates by providing Gemini with the scene-text annotation of a scene, and asking it for the 20 templates most relevant to that scene. From these, we randomly se-

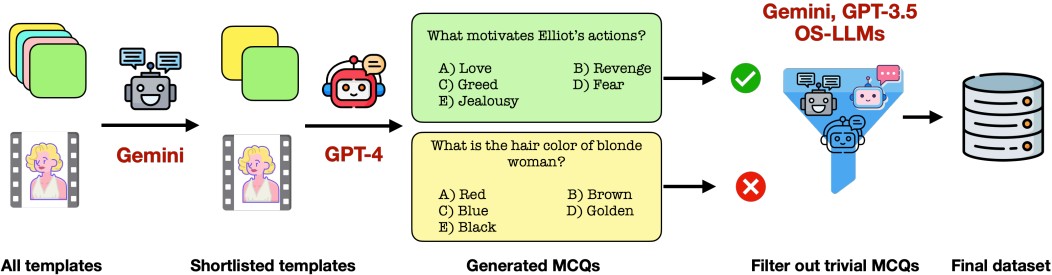

Figure 2. **Automated QA Generation and Filtering** Our process begins with a set of automated templates and scenes. Initially, we filter out the templates relevant to each scene. Next, we pass these templates along with the annotated-scene-text to GPT-4, which is then used to create multiple-choice questions (MCQs). The generated MCQs are then subjected to numerous filters to curate the final dataset. For more detailed information, refer to Sec. 2.3 and Sec. 2.4

lect 5-6 templates. We provide a commercial language model with (i) the audio description of a scene, which includes both visual descriptions and dialog, (ii) the selected question template names (e.g. 'Physical Possession'), (iii) the prototypical questions for the templates (e.g. "What is [Character Name] holding"), and (iv) a system prompt to generate questions. (complete pipeline shown in Fig. 2)

Through rigorous experimentation, we devised a system prompt that makes the model attentive to the entire scene and is capable of generating deeper, longer-term questions as opposed to mere surface-level perceptual queries. We observed that providing the prototypical example prevents GPT-4 from hallucination, and also leads to more plausible multiple-choice question (MCQ) distractors. We also found that asking the model to provide rationale for its answer enhances the quality of the questions. Additionally, we found that including timestamps for dialogues and visual descriptions augments the quality of generated temporal questions. We were able to generate ≈ 26 questions for each video in the dataset. While GPT-4 performs well across all question templates, we found that Gemini excels particularly with perceptual templates. Therefore, we utilized Gemini to generate a segment of questions in the dataset, while using GPT-4 for reasoning templates.

A small proportion of questions produced can be answered directly i.e., without referring to the clip, such as `What's the color of the blonde woman's hair?`. We implemented a few checks to eliminate trivial or poorly framed questions. We discuss these checks and a few axes we evaluate the question-answering dataset next.

### 2.4. Testing the quality of the dataset

While the process above consistently produces well-formed and answerable questions, we observed that some questions are either trivial, with answers embedded within the question itself, or pertaining to basic world concepts that do not require viewing the clip. To prune these, we evaluated our dataset with the help of a few LLMs on the following axes

and we either removed the questions from the dataset or compute metrics that users can use in the downstream tasks.

**Degeneracy.** A question is considered degenerate if the answer is implicit in the question itself, e.g., `What is the color of the pink house?`. In our dataset, these types of questions constitute only a small fraction. Manually reviewing all questions being impractical, we employed three distinct language models (LMs) to automate this process: Gemini [26], GPT-3.5 [1], and Phi-1.5 [13]. These models vary in their underlying training data and sizes. We presented only the questions and choices to these models, omitting any context, and calculated the accuracy of each question across the multiple models. If all models correctly answer a question, it is likely to be degenerate. We excluded degenerate questions from CinePile's evaluation split.

**Vision Reliance.** When generating the multiple-choice questions (MCQs), we considered the entire scene without differentiating between visual text and dialogue. Consequently, some questions in the dataset might be answerable solely based on dialogue, without needing the video component. For this analysis, we utilized the Gemini model. The model was provided with only the dialogue, excluding any visual descriptions, to assess its performance. If the model correctly answers a question, it is assigned a score of 0 for the visual dependence metric; if it fails, the score is set at 1. In later sections, we present the distribution of the visual dependence scores across different MCQ categories.

**Hardness.** We developed a metric to gauge the difficulty of questions for the models, even when provided with full context. For this purpose, we selected the Gemini model, given its status as one of the larger and more capable models. This metric differs from accuracy; during evaluation, the models are only supplied with videos and dialogue information, excluding visual descriptions. However, in calculating the hardness metric, we include visual descriptions as part of the context given to the model.

Additionally, authors regularly verified the quality of questions across multiple scenes and corrected any systemic

errors that arose in the pipeline. We also conducted a human study to identify weaknesses in the dataset, further discussed in Appendix Sec. L.

## 3. A look at the dataset

Our dataset consists of 8000 video clips with average length of ≈160 seconds, split into train and test splits of 7700 and 300 videos each. Following the pipeline outlined in Sec. 2, we ended up with over 200,000 training points and 7,800 test-set points (before degeneracy filtration). Each MCQ contains a question, answer, and four distractors. After filtration of the degenerate questions from the test split, we are left with 5,500 questions. Of all the test questions, 34.30% are reliant on visual information. We present additional dataset statistics including distribution of questions, hardness scores across different categories, in Appendix Sec. G.

## 4. Model evaluation

In this section we discuss the evaluations of various closed and open source video LLMs on our dataset, some challenges, and the model performance trends. Given that our dataset is of type multiple-choice question answers (MCQs), we evaluate a given model's performance on our benchmark questions by measuring its ability to accurately select the right answer from a set of options, containing only one correct answer and four distractors. One key challenge is reliably parsing the model's response to extract its chosen answer, and mapping it to one of the predefined answer choices. Model's responses may vary in format, including additional markers, or may only contain the option letter, or have a combination of the option letter and its corresponding text, etc. Such variations necessitate a robust post-processing step to accurately extract and match the model's response to the correct answer. Due to space constraints, we discuss the process in detail in Appendix Sec. H.

During the evaluation, we specifically instruct the model to be concise and only output the option letter. Qualitatively we see that most commercial models are good at following these instructions, and we can map these responses well. Some OSS models are very verbose in their response, and poor at following instructions. Hence, we also computed traditional video-caption evaluation metrics like BertScore [42], CIDEr [28], and ROUGE-L [14] for open-source models, and present results in Appendix.

We evaluate various commercial and open-source LLM models and we present their performance in Tab. 1. We also present human numbers (author and non-author) for comparison. On average, VLM models both commercial and OSS, are behind human performance on our dataset. While commercial VLMs perform reasonably well, the OSS models perform quite poorly showing the gap in their capabilities. Among the question categories, GPT-4V model

Table 1. **Evaluations on CinePile.** Accuracy of various video LLMs on the test split, along with Human performance for comparison. Chance performance is 20%. TEMP refers to Temporal. See Sec. 2.2 for other acronyms.

| Model | Average | CRD | NPA | TEMP | STA | TH |
|---|---|---|---|---|---|---|
| Human | 73.21 | 82.92 | 75.00 | 75.52 | 73.00 | 64.93 |
| Human (authors) | **86.00** | **92.00** | **87.5** | **100** | 71.20 | **75.00** |
| GPT-4 Vision [1] | 66.75 | 68.14 | 76.54 | 65.33 | **76.04** | 57.19 |
| Gemini Pro Vision [26] | 57.68 | 59.25 | 70.37 | 56.80 | 63.54 | 46.15 |
| Claude 3 (Opus) [2] | 46.34 | 46.15 | 64.19 | 44.82 | 47.91 | 39.46 |
| mPLUG-Owl [37] | 15.93 | 15.92 | 14.19 | 14.88 | 15.78 | 18.85 |
| Video-ChatGPT [16] | 14.70 | 16.44 | 11.72 | 11.83 | 29.03 | 12.20 |
| MovieChat [23] | 7.12 | 6.96 | 6.78 | 8.05 | 11.32 | 5.08 |

performed poorest in "Thematic Exploration" category followed by "Temporal". Surprisingly, GPT-4V did well in "Setting and Technical Analysis" which pertains to purely visual questions. One possible reason for this is, GPT-4V is aware of famous movies and it can answer the questions even without any context. To understand the extent of this memorization, we computed the degeneracy metrics for the commercial models, we see GPT-4 stands at 33.4%, Gemini at 39.64% and Claude at 34%. This states that the true performance of these models might be much lower than the numbers reflected in the table! However, we hope that training OSS models on our data will help them match the performance of commercial models in the future.

## 5. Discussion and Conclusion.

In this paper, we introduced CinePile, a unique long video understanding dataset and benchmark, featuring over 200k questions in the training set and 5500 in the test split. We detailed a novel method for curating and filtering this dataset, which is both scalable and cost-effective. Additionally, we benchmarked various recent commercial multimodal LLMs and conducted a human study to gauge the achievable performance on this dataset. To our knowledge, CinePile is the only large-scale dataset that focuses on multi-modal understanding, as opposed to the purely visual reasoning addressed in previous datasets. We intend to make the questions and answers from the training set of CinePile publicly available. Additionally, we will set up a leaderboard for the test set, providing a platform for new video LLMs to assess and benchmark their performance on CinePile.

Despite its strengths, there are still a few areas for improvement in our dataset, such as the incorporation of character grounding in time. While we believe our dataset's quality is comparable to or even better than that of a Mechanical Turk annotator, we acknowledge that a motivated human, given sufficient time, can create more challenging questions than those currently generated by an LLM. Our goal is to narrow this gap in future iterations of CinePile.

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
