# OpenReview forum: "CinePile: A Long Video Question Answering Dataset and Benchmark"
_thecvf.com/CVPR/2024/Workshop/SyntaGen — SyntaGen 2024_

### Official Review · Reviewer_S3Mt · 2024-03-27
**This paper present a new benchmark for authentic long-form video understanding, which utilizing LLMs to generate  200,000 multiple-choice questions.**

**Rating:** 6
**Confidence:** 4

**Review:**

The main concerns raised by some reviewers are as follows:

1. The reviewer recommends organizing the experimental conclusions from Section 3 Human Study into corresponding figures or tables, as this would aid comprehension. This section of the experiment is crucial for evaluating the quality of captions generated by LLM.

2. Some relevant works are missing [1][2], which also explore the effectiveness of using LLM to generate captions. Although they focus on the image level, discussing and comparing them in the paper would seem more reasonable.

3. "L341"authors have independently taken the survey, and the corresponding accuracy is 86%". what's meaning for the survey. It looks like there haven't been any relevant definitions provided.

[1] Paragraph-to-image generation with information-enriched diffusion model

[2] PixArt: Fast Training of Diffusion Transformer for Photorealistic Text-to-Image Synthesis

---

### Official Review · Reviewer_tQuB · 2024-03-31
**Review of the paper.**

**Rating:** 6
**Confidence:** 3

**Review:**

Presentation: The paper doesn't update the submission number while building the PDF.

This work proposes a pipeline to build a video-QA dataset, which finally delivers a large-scale dataset of 8,000 video clips, accompanied with around 200,000 template-based questions. The uniqueness of the dataset lies in the long video natural (average 160 seconds per video) and focuses on temporal reasoning and video understanding rather than solely visual questions. The benchmark dataset illustrates that current LLMs still have room for improvement, and the human performances are also quite reasonable, given that the questions are still error-prone in some cases due to the ambiguity in both the questions and answer choices.

However, one aspect that the authors didn't illustrate the improvement upon LLMs when fine-tuning with the proposed training dataset, which takes ~96% of the overall proposed dataset. It would be more convincing if the training dataset also is effective or not.

---

### Official Review · Reviewer_NVy8 · 2024-04-03

**Rating:** 7
**Confidence:** 3

**Review:**

This paper introduces a dataset for long-form video understanding. Unlike previous datasets, the QA problem doesn't focus much on the clues gathered from static appearance information. This increases the challenge of the benchmark. Additionally, the proposed dataset consists of a significant amount of data, powered by an introduced automatic pipeline, which enables downstream instruction tuning. The pros of this work contribute to a potentially large benchmark for long-video understanding. There are no apparent cons regarding the contribution.

---

### Decision · Program_Chairs · 2024-04-06

**Decision:**

Accept

**Comment:**

All reviewers recommended acceptance (1 accept, 2 borderline accept) and the paper is generally well-received. A key strength of this paper is its introduction of a novel, large-scale dataset designed specifically for long-form video understanding.
The PCs agreed with the recommendation. However, there are several areas where revisions are recommended. These include providing more explicit evidence of the dataset's effectiveness, improving the presentation of experimental results, and expanding the literature review to include missing related works.
Please incorporate the feedback into the revision. Congratulations!